# Use of Biotechnological Methods to Support the Production of New Peach Hybrids

**Irina Mitrofanova *** , **Nina Lesnikova-Sedoshenko** , **Valentina Tsiupka, Anatoliy Smykov**
**and Olga Mitrofanova**

Plant Developmental Biology, Biotechnology and Biosafety Department, Federal State Funded Institution of
Science "The Nikita Botanical Gardens–National Scientific Center of the RAS", 298648 Yalta, Russia;
nplesnikova@yandex.ru (N.L.-S.); valentina.brailko@yandex.ru (V.T.); selectfruit@yandex.ru (A.S.);
invitro_plant@mail.ru (O.M.)
**\* Correspondence: irimitrofanova@yandex.ru

**Abstract:** Peach [*Prunus persica* (L.) Batsch] is among the most demanded fruit crops in the world. Biotechnological methods help to originate new hybrid forms in order to increase the cultivar diversity and create new valuable genotypes. Cross combinations between the cultivars Clyde Wilson, Jerseyglo, Loadel, Summerglo and the promising cultivar 'Nikitskiy Podarok' have been done. The embryos of these hybrids germinated and formed plantlets after stratification at 4 °C for 45–60 days. The best regeneration rates in the hybrids 'Loadel' × 'Nikitskiy Podarok' and 'Summerglo' × 'Nikitskiy Podarok' (96.30% and 92.59%, respectively) were noted on hormone-free Monnier culture medium supplemented with 400.0 mg L$^{-1}$ casein hydrolyzate. When the newly formed plantlets had necrosis of the shoot apex or immature roots, nodal shoot segments were used. At the same time, a high regeneration capacity was noted in the hybrids 'Summerglo' × 'Nikitskiy Podarok' and 'Loadel' × 'Nikitskiy Podarok' on B5 culture medium with 0.75 mg L$^{-1}$ 6–benzyl–aminopurine (BAP) + 0.1 mg L$^{-1}$ indole–3–butyric acid (IBA). After the second subculture, the number of new adventitious shoots was $5.18 \pm 0.18$ and $4.95 \pm 0.18$ shoots per explant, respectively. The plants obtained from the hybrid embryos in a soil mixture soil: peat: sand (3:1:1) were adapted. The main morphological and anatomical features of the leaf blades in newly originated peach hybrids have been studied: the thickness of their tissues and the distribution of stomatal apparatus, as well as the physiological parameters of the photosystem II activity in regenerants cultured in vitro and during their in vivo acclimatization. The high capacity to post aseptic adaptation in the obtained hybrids has been shown.

**Keywords:** *Prunus persica* (L.) Batsch; embryoculture; plant growth regulators; regeneration capacity; leaf structure; mesophyll; stomata; photosynthetic activity

## 1. Introduction

Peach [*Prunus persica* (L.) Batsch, genus *Prunus* L., family Rosaceae Juss.] is one of the main economically important stone fruit crops. It has been grown in more than 60 countries all over the world, its orchards occupy more than 1.5 million hectares [1]. The leaders in the production of peach fruits are China, Iran, Italy, Spain, USA, Turkey, Greece, France [1,2]. In Russia, peaches are grown in the southern regions of the country [3,4]. Peach fruits are characterized by high dietary and medicinal properties, are suitable for fresh consumption and for processing, have a wide range of ripening and are in great demand on the market. Nowadays, viral diseases of stone fruit crops are a great problem for commercial horticulture, due to the widespread distribution of viral phytopathogens, such as *Plum pox Potyvirus* (PPV), *Prunus necrotic ringspot Ilarvirus* (PNRSV), *Prune dwarf Ilarvirus* (PDV), *Apple mosaic Ilarmeretirus* (ApMV) [5]. Viruses cause significant damage to fruit trees, reducing yields by 30–100%. The most dangerous and widespread on stone fruit crops is *Plum pox Potyvirus*, which reduce the yield and fruit quality, weaken the plants

and cause their death [6–10]. Nowadays, in a number of countries, breeding programs to obtain new cultivars and forms of stone fruit crops have been developed. The peach gene pool collection at the Nikita Botanical Gardens has been formed since its foundation in 1812. At present, the collection includes 717 cultivars and breeding forms and a wide range of genotypes from various ecological-geographical groups, which makes it possible to carry out effective research on the originating of new cultivars and hybrids with a complex of economically valuable traits. Our studies included cultivars close in their origin. The cultivar Nikitsky Podarok was originated as a result of crossing the Canadian cultivar Veteran and the American one–Cardinal. Cultivars Clyde Wilson, Jerseyglo, Loadel, Summerglo were bred in the USA by hybridization between the American cultivars. At the same time, the cultivars Clyde Wilson and Jerseyglo belong to the Northern Chinese ecological-geographical group, and Nikitsky Podarok, Loadel, Summerglo belong to the Iranian group. The cultivars are characterized by a complex of biologically valuable features. Thus, the pomological description of the cultivars showed that three cultivars are large-fruited (165.0–185.0 g): Clyde Wilson, Jerseyglo, Summerglo. The other cultivars had a medium fruit weight. All cultivars had the fibrous flesh, with the exception of Loadel, in which the flesh is gristly. The fruits of the studied cultivars were characterized by good appearance (4.3–4.5 points) and harmonious taste (4.4–4.5 points), especially in the Summerglo cultivar (4.6 points). Their frost and drought tolerance is medium, except the cultivar Nikitsky Podarok, in which a high drought tolerance demonstrated. The cultivars Nikitsky Podarok and Clyde Wilson give medium yield and in the cultivars Jerseyglo, Loadel and Summerglo, high yield. In the cultivar Nikitsky Podarok during the last 15 years, *Plum pox virus* tolerance was demonstrated.

One of the ways to obtain new hybrids is through biotechnological methods, including in vitro embryoculture. However, under subsequent stages the micropropagation technique is requested due to germs having undeveloped epicotyl or hypocotyl during in vitro embryo development, and not forming full seedlings.

The ability of plants to adapt to growing conditions is one of the most important criteria in assessing the prospects of their use for various purposes (harvesting, breeding, and commercial cultivation), and during the development of zonal assortments of useful plants [11]. In turn, the mechanisms of species, forms and cultivars adaptation to environmental conditions are determined by their anatomical, morphological, physiological and biochemical structural features. The acclimatization process, that is, the transfer of a plant from in vitro to ex vitro conditions, is often a critical stage for a seedling survival [12–16]. There are a number of differences between in vitro and ex vitro conditions in terms of light intensity and quality, relative air humidity, availability of nutrients, substrates and gas concentration. These factors can cause disturbances in growth, development, morphology and physiology of plants in vitro [14,17,18]. Among the main causes of functional disorders in plants ex vitro are the imperfect functioning of the stomatal apparatus, poor development of the cuticle and non-functioning root system [13,19,20]. In this regard, some authors reasonably recommend the use of structural changes monitoring in the vegetative organs of investigated plants adapted ex vitro for the development and optimization of the clonal micropropagation protocol [14,21–26], as well as to study their functional abilities [27,28].

The objective of our study was to obtain by using biotechnological methods new hybrids of peach for the further creation, preservation, propagation of valuable genotypes and to characterize and evaluate changes in the structure and functions of the leaves in new peach hybrids cultured in vitro and during their ex vitro acclimatization.

## 2. Materials and Methods

### 2.1. Plant Materials

Research was done in the Plant Biotechnology and Virology Laboratory of the Plant Developmental Biology, Biotechnology and Biosafety Department of the Federal State Funded Institution of Science "The Nikita Botanical Gardens–National Scientific Center of the RAS" (FSFIS "NBG-NSC"). The initial plant material for in vitro culture was

hybrid seeds obtained in 2019 and 2020 as the result of the cross-pollination of peach cultivars [*Prunus persica* (L.) Batch], previously tested for the absence of phytoviruses. In the experiments we used hybrid embryos of four cross combinations 'Clyde Wilson' × 'Nikitskiy Podarok', 'Jerseyglo' × 'Nikitskiy Podarok', 'Loadel' × 'Nikitskiy Podarok' and 'Summerglo' × 'Nikitskiy Podarok'.

The main characteristics of the cultivars are:

Nikitsky Podarok. The cultivar was originated in the Nikita Botanical Gardens by Z.N. Perfilieva, A.V. Smykov, V.K. Smykov, O.S. Fedorova as a result of crossing the cultivars Veteran and Cardinal. It was listed in the Register of Breeding Achievements of the Russian Federation, accepted for use. Flowers are bell-shaped. Fruits are medium size (125 g), round with carmine blush (75–100% of the fruit surface) on a yellow background. Flesh is yellow, fibrous, juicy with harmonious taste (score 4.5 points), clingstone. Fruit yield 127 kg/ha. The fruits ripen in the third decade of July.

Clyde Wilson. The cultivar was originated in 1981 by William J. Wilson in Fort Valley Georgia, the USA as a result of a mutation in Loring cultivar. It was introduced to the Nikita Botanical Gardens in 1987. The flower is rose-shaped. Fruits are round, medium-sized (95–130 g) with a carmine blush (50–75% of the fruit surface) on the yellow background. The flesh is yellow, fibrous, freestone, the taste is 4.4 points. The fruit ripens in the 2–3 decades of August.

Jerseyglo. The cultivar was originated in 1979 by C. Bailey, L.F. Hough at New Jersey Agricultural Experiment Station, New Brunswick, USA by crossing the cultivars Jefferson × Loring. It was introduced to the Nikita Botanical Gardens in 1983. The flower is rose-shaped. Fruits are round, medium-sized (120 g) with a red blush (10–25%) on the yellow background. The flesh is yellow, fibrous, freestone, the taste is 4.5 points. The fruit ripens in the third decade of August–first decade of September.

Loadel. The cultivar was originated in 1950 by Howard H. Harter in Yuba City, California, USA from free pollination of the cultivar Lovell. It was introduced to the Nikita Botanical Gardens in 1975. Bell-shaped flower. Fruits are round, medium-sized (115 g) with carmine blush (25–50% of the fruit surface) on the yellow background. The flesh is firm, yellow, gristly, freestone, the taste is 4.3 points. The fruit ripens in the 2–3 decades of August.

Summerglo. The cultivar was originated in 1978 by C. Bailey, L.F. Hough at New Jersey Agricultural Experiment Station, New Brunswick, USA by crossing 'Collins' × 'Red Slovenia'. It was introduced to the Nikita Botanical Gardens in 1983. Bell-shaped flower. Fruits are round, large (150–180 g) with carmine blush (25–50% of the fruit surface) on the yellow background. The flesh is yellow, fibrous, clingstone, the taste is 4.2 points. The fruit ripens in the second decade of August.

### 2.2. Establishment of Embryoculture In Vitro

In order to introduce in vitro hybrid peach embryos and to regenerate seedlings from them, the biotechnology and in vitro embryoculture methods were used [29–31]. After removing from peach fruits exo-and mesocarpies stones were obtained. The stones by immersing them in 96% ethanol for 1–2 s, followed by firing in the flame of an alcohol lamp to sterilize. The stones were opened with a specially designed device to destroy the stony endocarp and the seeds were isolated. If the endocarp was cracked without external signs of damage by phytopathogens and insects, the seeds were sequentially sterilized in 70% ethanol (2 min), 1% Thimerosal solution (10 min), 0.4% chlorine DesTab solution (10 min) with 2–3 drops of Tween 20. After each reagent, the explants were rinsed 3–4 times in sterile distilled water. The embryos were isolated by removing the seed covers and placed on culture media under aseptic conditions in the SC2 Biological Safety Cabinet (ESCO, Singapore). To induce the germination of mature embryos and the formation of seedlings, a hormone-free Monnier culture medium [32] with 2.5% sucrose and 0.9% agar was used. When the immature embryos were germinated, culture media were supplemented with 0.1–0.5 mg L$^{-1}$ 6–furfurylaminopurine (kinetin), 1.0–2.0 mg L$^{-1}$ L–glutamine, 3.0 mg L$^{-1}$

glycine, 0.2–1.0 mg L$^{-1}$ gibberellic acid (GA$_3$), 250.0–400.0 mg L$^{-1}$ casein hydrolysate. The medium pH was adjusted to 5.7 before autoclaving in a sterilizer LAC 5060S (DAINAN LABTECH, Korea) at 120 °C for 15 min, depending on the volume of the culture vessel. Solutions of plant growth regulators (PGRs), vitamins, and ribavirin were sterilized by cold filtration through MILLEX® GP filters (0.22 μm) (Millipore, Germany) and added to culture media after autoclaving. The test tubes with embryos were placed in the dark in a refrigerator (HAIER HXC-608, China) under a low positive temperature of 4 ± 1 °C. On 45 and 60 days of in vitro culture at 4 °C the shoot and root average length, and the leaf number per explants were recorded. A total 20 embryos per treatment were studied. After 45–60 days, the formed plants were placed in a phytocapsule and a growth chamber with modeling climatic conditions MLR-352-PE (Panasonic, Japan), first with an average daily temperature of 15 ± 1 °C, then at 24 ± 1 °C, a 16–hour photoperiod, and light intensity 37.5 μmol m$^{-2}$ s$^{-1}$ provided with white fluorescent lamps (Philips TL, 40W, Japan).

The effect of genotype and embryo size on the plantlets formation from hybrid peach embryos on Monnier culture medium on 90 days after initial culture was investigated. Finally, the regeneration frequency of hybrid embryos on culture medium without PGRs and on medium with 0.4 mg L$^{-1}$ Kinetin + 0.1 mg L$^{-1}$ GA$_3$ plants was recorded. A total 20 embryo per treatment were investigated. Plant objects were examined using a Nikon SMZ745T binocular microscope (Japan) equipped with an Industrial Digital Camera 5.1 MP 1/25 Color USB2.0 (China) with the appropriate software.

### 2.3. Adventitious Shoot Regeneration

To induce adventitious shoot formation, shoot segments (1–2 nodes) of the plantlets with undeveloped roots or necrosis of the apical part were used. The culture media were composed in accordance with the stages of plant morphogenesis, taking into account the ratio of vitamins, carbohydrates and PGRs. The morphogenetic and regenerative capacity of peach seedlings were assessed on the media: B5 [33] supplemented with 0.5, 0.75, 1.0 mg L$^{-1}$ 6–benzyl–aminopurine (BAP) and 0.1–0.2 mg L$^{-1}$ indole–3–butyric acid (IBA); MS [34] with 1.27 mg L$^{-1}$ [1–phenyl–3–(1,2,3–thiadiazol–5–yl) urea] (TDZ, thidiazuron); MS with 1.0–2.0 mg L$^{-1}$ BAP + 1.0–2.0 mg L$^{-1}$ indole-3-acetic acid (IAA). At the induction stage of shoot segments development, the main characteristic of regeneration processes was the number of newly formed adventitious shoots in vitro. The trials were performed during four weeks to compare a two culture medium (MS and B5) with and without RGRs for shoot formation per explant from the nodal segments of four peach hybrids. The trials were repeated twice in 20 replications using shoot segments cultured in culture vessels with solid medium for multiple adventitious shoot induction.

### 2.4. Acclimatization of Plantlets

Plantlets obtained from the hybrid embryos in vitro, with a well-formed root system, were transferred for adaptation into plastic pots 7.5 cm diameter with a mixture of soil: peat: perlite (3:1:1), covered with insulators and grown in a plant growth climatic chamber (Conviron, Canada) at 24 ± 1 °C, a light intensity 37.5 μmol m$^{-2}$ s$^{-1}$, a photoperiod of 16 h and air humidity 70–80%. Acclimatization was gradual: two months ex vitro in a plant growth chamber, then under greenhouse conditions.

### 2.5. Leaf Morphological and Anatomical Examination

To describe the morphological features of the leaves from microshoots, 5–10 plants were selected at each stage of the culture and adaptation: from plantlets in vitro, with adaptation for 15 days in a plant growth chamber Conviron and after 60 days culture in a greenhouse. The rooted plants from the growth chamber to the greenhouse after 30 days of adaptation were removed. To study the structure, the middle part of the leaf was excised and frozen on a microtome (MZ–2, Ukraine) equipped with an OL–ZSO 30 laboratory cooling system (Inmedprom, Russian Federation). The sections were stained with a 1% solution of phloroglucinol, methylene blue (to stain the vascular system), and

then placed in a glycerol solution [35]. The samples were examined using a CX41 light microscope (Olympus, Japan) equipped with an SC 50 camera (Olympus, Germany) and CellSensImaging Software version 1.17. The sections were used to determine the thickness of the leaf, epidermis, cuticle and mesophyll tissues, as well as the size and number of stomata per 1 mm$^2$ of abaxial epidermal tissues.

### 2.6. Methods for Assessing the Functional State of Plants

Investigations of the pigment apparatus in leaves from the peach microshoots cultured in vitro and adapted plants in vivo (15 and 60 days) were made with a portable pulse fluorometer MINI-PAM II, WALZ (HeinzWalz GmbH, Germany) according to the methodological recommendations by Stirbet and Govindjee [36]. Before measuring the fluorescence parameters, the leaves were adapted to the dark for 30 min. The parameters of the photoinduction curve recorded in the experiments ($F_m$–maximum fluorescence, $F_t$–stationary level of fluorescence, $F_m'$–maximum level of fluorescence in light) made it possible to calculate the main indexes of the photosynthetic apparatus: photosynthetic activity PA = $(F_m - F_t)/F_m$ and effective photochemical quantum yield of PS II–Y (II) = $(F_m' - F_t)/F_m'$.

Chlorophyll amount in tissues was determined with a non-destructive optical sensor DUALEX (FORCE-A, Orsay, France) [37,38]. Due to chemical calibration by FORCE-A, data for chlorophyll amount is presented in µg cm$^{-2}$ in the range of 5–80 µg cm$^{-2}$.

### 2.7. Statistical Analysis

The experiments were carried out twice in twenty replications. Statistical analyses of the obtained data were made by STATISTICA for Windows 10.0 (StatSoft, Inc., Tulsa, OK, USA) and Duncan's multi-rank test ($p \leq 0.05$).

## 3. Results

### 3.1. Embryoculture

The genotype, the size of the embryo, the culture medium, the plant growth regulators (PGRs) in the medium, the duration of stratification at positive low temperatures significantly affected the peach hybrid embryo germination and plant formation. Some morphometric parameters of embryo germination of the studied peach cross combinations, depending on the stratification duration and genotype, are presented in Table 1.

**Table 1.** Germination of hybrid peach embryos on Monnier culture medium, depending on the duration of stratification at 4 °C.

| Genotype | Days after Initial Culture | | | | | |
| | 45 | | | 60 | | |
| | Length (cm) * | | No. Leaves per Explant * | Length (cm) * | | No. Leaves per Explant |
| | Shoot | Root | | Shoot | Root | |
| 'Clyde Wilson' × 'Nikitskiy Podarok' | 0 [b,c] | 0.82 ± 0.07 [b,c] | 0 [c] | 1.78 ± 0.18 [b,c] | 1.71 ± 0.18 [b,c] | 2.35 ± 0.31 [b,c] |
| 'Jerseyglo' × 'Nikitskiy Podarok' | 0.68 ± 0.07 [a] | 0.99 ± 0.30 [a,b] | 0.17 ± 0.09 [a,b] | 2.66 ± 0.28 [a,b] | 3.09 ± 0.30 [a,b] | 2.85 ± 0.18 [a,b] |
| 'Loadel' × 'Nikitskiy Podarok' | 0.89 ± 0.06 [a] | 1.34 ± 0.12 [a] | 0.35 ± 0.13 [a] | 3.06 ± 0.23 [a,b] | 4.42 ± 0.59 [a] | 3.60 ± 0.29 [a] |
| 'Summerglo' × 'Nikitskiy Podarok' | 0.52 ± 0.07 [a,b] | 0.99 ± 0.14 [a,b] | 0.15 ± 0.11 [a,b] | 2.68 ± 0.25 [a] | 3.54 ± 0.29 [a,b] | 3.20 ± 0.19 [a] |

* Different lowercase letters in the same column indicated the significant difference at $p \leq 0.05$ (Duncan's multiple-range test).

Our previous studies on the isolated peach embryos culture revealed that Monnier medium was the most favorable one for the development of peach embryos [39]. The germination of hybrid embryos and further regeneration of plantlets in the studied peach

cross combinations were induced on Monnier medium (Figure 1, Table 2). Cold pretreatment of hybrid peach embryos was 45–60 days. In this case, the development of normal seedlings was observed (Figure 2). In the absence of cold treatment, abnormal development of seedlings was noted.

Embryos 'Loadel' × 'Nikitskiy Podarok' and 'Summerglo' × 'Nikitskiy Podarok' produced more plantlets on both hormone-free Monnier medium (96.30 and 92.59%, respectively) and medium with 0.4 mg L$^{-1}$ Kinetin + 0.1 mg L$^{-1}$ GA$_3$. The lowest frequency of shoot regeneration from embryos 0.3–1.0 cm long was noted in hybrid embryos 'Clyde Wilson' × 'NikitskiyPodarok' (3.70%) and 'Jerseyglo' × 'Nikitskiy Podarok' (7.41%) on hormone-free Monnier medium.

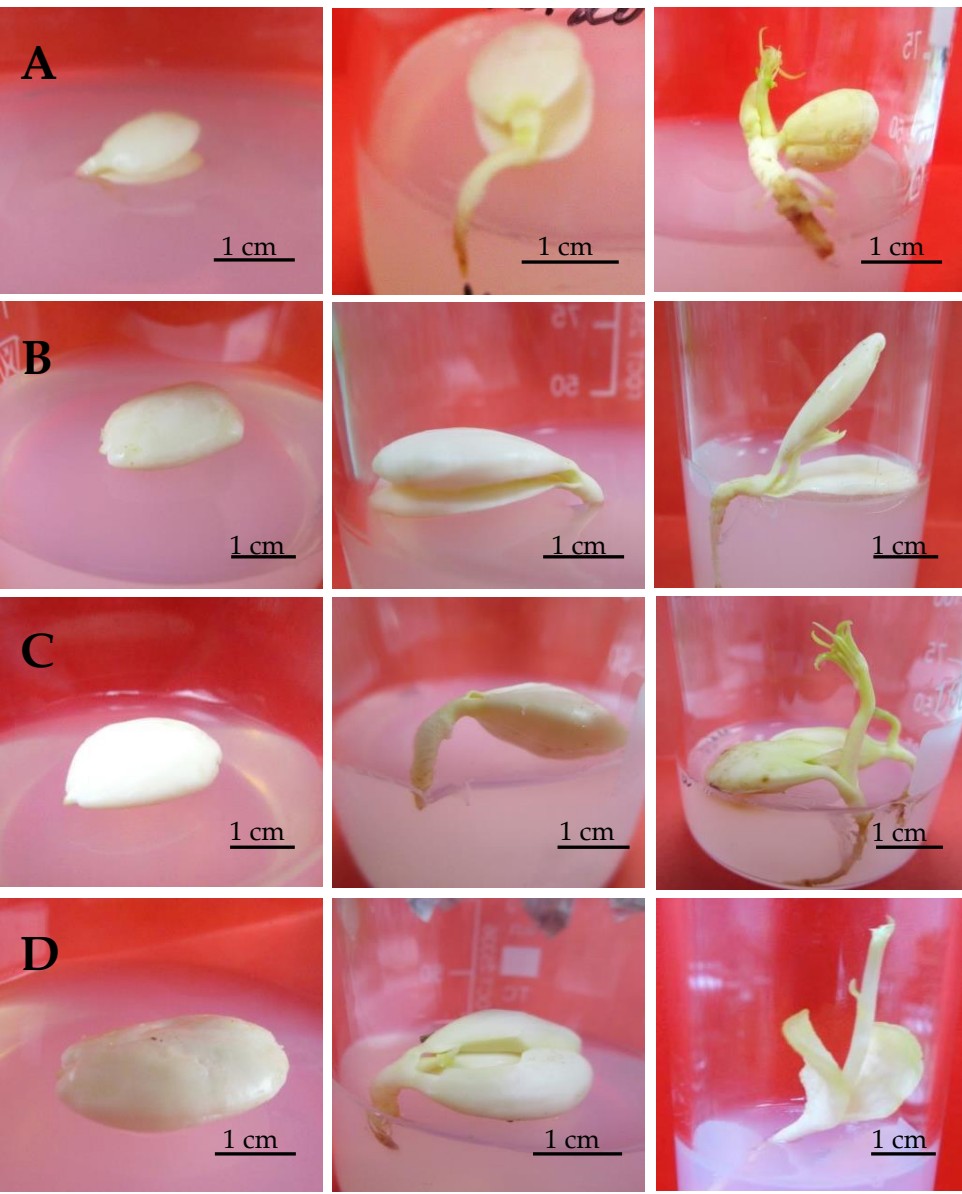

**Figure 1.** Germination of peach hybrid embryos on Monnier culture medium: (**A**) 'Clyde Wilson' × 'Nikitskiy Podarok', (**B**) 'Jerseyglo' × 'Nikitskiy Podarok', (**C**) 'Loadel' × 'Nikitskiy Podarok', (**D**) 'Summerglo' × 'Nikitskiy Podarok'.

**Table 2.** Effect of genotype and embryo size on the plantlets formation from hybrid peach embryos on Monnier culture medium (90 days after initial culture).

| Genotype | Length of the Embryo, cm | Regeneration Frequency *, % | |
|---|---|---|---|
| | | Culture Medium without PGRs | Culture Medium with 0.4 mg L$^{-1}$ Kinetin and 0.1 mg L$^{-1}$ GA$_3$ |
| 'Clyde Wilson' × 'Nikitskiy Podarok' | 0.3–1.0 | 3.70 $^{e,f}$ | 11.11 $^{e,f}$ |
| | 1.1–2.0 | 62.96 $^{a,b}$ | 51.85 $^{a,b}$ |
| 'Jerseyglo' × 'Nikitskiy Podarok' | 0.3–1.0 | 7.41 $^{c,d}$ | 14.81 $^{c,d}$ |
| | 1.1–2.0 | 85.19 $^{a,b}$ | 74.07 $^{a,b}$ |
| 'Loadel' × 'Nikitskiy Podarok' | 0.3–1.0 | 18.51 $^{b,c}$ | 29.62 $^{a,b}$ |
| | 1.1–2.0 | 96.30 $^{a}$ | 88.89 $^{a}$ |
| 'Summerglo' × 'Nikitskiy Podarok' | 0.3–1.0 | 14.81 $^{bc}$ | 22.22 $^{bc}$ |
| | 1.1–2.0 | 92.59 $^{a}$ | 81.48 $^{a}$ |

* Different lowercase letters in the same column indicated the significant difference at $p \leq 0.05$ (Duncan's multiple-range test).

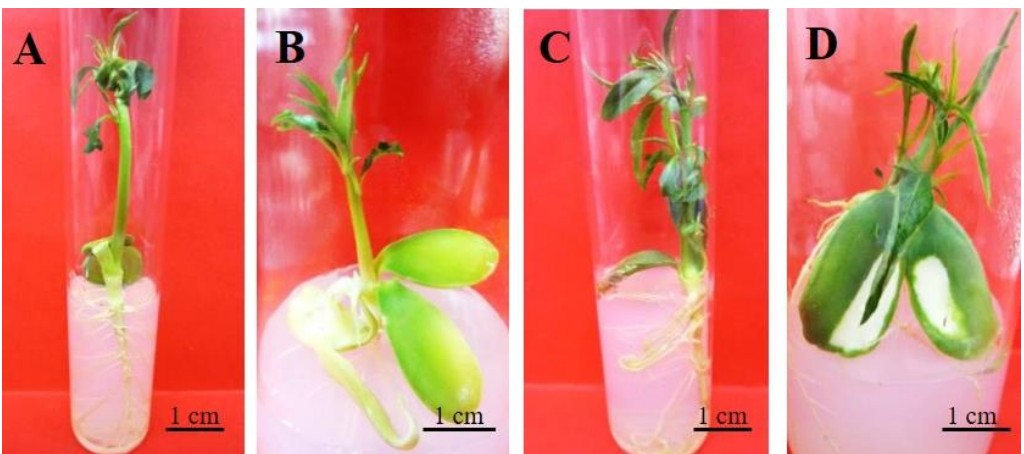

**Figure 2.** Plantlet regeneration from hybrid embryo in *Prunus persica*: (**A**) 'Clyde Wilson' × 'Nikitskiy Podarok', (**B**) 'Jerseyglo' × 'Nikitskiy Podarok', (**C**) 'Loadel' × 'Nikitskiy Podarok', (**D**) 'Summerglo' × 'Nikitskiy Podarok'.

*3.2. Shoot Regeneration*

All tested combinations of plant growth regulators in culture media B5 and MS activated the process of morphogenesis and induced shoot formation from axillary buds in the studied peach hybrids, compared to the control (Table 3). Shoots of the cross combinations 'Summerglo' × 'Nikitskiy Podarok' and 'Loadel' × 'Nikitskiy Podarok' demonstrated a high regeneration capacity. After the second subculture on B5 medium with 0.75 mg L$^{-1}$ BAP + 0.1 mg L$^{-1}$ IBA, the number of newly formed adventitious shoots was 5.18 ± 0.18 and 4.95 ± 0.18 shoots per explant, respectively. In the treatments with 1.5 mg L$^{-1}$ BAP and 1.5 mg L$^{-1}$ IAA in MS medium, this indicator was a little lower (4.79 ± 0.29 and 4.33 ± 0.13, respectively). However, when using 1.27 mg L$^{-1}$ TDZ and a combination of 1.5 mg L$^{-1}$ BAP + 1.5 mg L$^{-1}$ IAA, callus formed at the base of the explants (Figure 3). The research on the obtained shoot rooting in vitro has been started.

*3.3. Plantlet Acclimatization*

In vitro produced plantlets of peach hybrids for adaptation to ex vitro conditions were transferred. It is well known that to transfer plantlets to greenhouse and then to open field requires a preadaptation period. Firstly, plantlets with a well-formed root system were planted in a mixture of soil: peat: perlite (3:1:1), covered with isolators, and grown in a controlled climate chamber for plant growth. Under these conditions, the plants adapted for 3–4 weeks. After ex vitro acclimatization, seedlings were placed in an insulated nursery

rack. After 2 months, the seedlings were transferred into larger containers with a substrate mixture–soil: peat: sand (3:1:1). Figure 4 presents the adapted peach hybrids.

**Table 3.** Effect of the culture media, different plant growth regulators and their concentrations on the shoot formation per explant from the nodal segments of four hybrids of the peach.

| Culture Media | PGRs (mg L$^{-1}$) | | | | 'Clyde Wilson' × 'Nikitskiy Podarok' * | 'Jerseyglo' × 'Nikitskiy Podarok' * | 'Loadel' × 'Nikitskiy Podarok' * | 'Summerglo' × 'Nikitskiy Podarok' * |
|---|---|---|---|---|---|---|---|---|
| | BAP | TDZ | IBA | IAA | | | | |
| B5 (control) | 0 | 0 | 0 | 0 | 0.38 ± 0.12 [c] | 0.51 ± 0.11 [b] | 0.72 ± 0.12 [a] | 1.08 ± 0.12 [a] |
| | 0.5 | 0 | 0.1 | 0 | 2.08 ± 0.15 [c] | 3.25 ± 0.09 [a] | 3.13 ± 0.23 [a,b] | 3.67 ± 0.20 [a] |
| | 0.5 | 0 | 0.2 | 0 | 2.21 ± 0.13 [c] | 3.18 ± 0.09 [a] | 2.46 ± 0.19 [a,b] | 4.18 ± 0.13 [a] |
| B5 | 0.75 | 0 | 0.1 | 0 | 3.03 ± 0.09 [b,c] | 3.72 ± 0.18 [a,b] | 4.95 ± 0.18 [a] | 5.18 ± 0.18 [a] |
| | 0.75 | 0 | 0.2 | 0 | 2.87 ± 0.12 [c,d] | 3.43 ± 0.09 [b] | 4.03 ± 0.11 [a] | 4.64 ± 0.19 [a] |
| | 1.0 | 0 | 0.1 | 0 | 2.79 ± 0.11 [c] | 3.79 ± 0.16 [a] | 3.18 ± 0.26 [a,b] | 4.31 ± 0.22 [a] |
| | 1.0 | 0 | 0.2 | 0 | 2.51 ± 0.12 [bc] | 2.95 ± 0.27 [a,b] | 3.59 ± 0.19 [a] | 3.54 ± 0.19 [a] |
| MS (control) | 0 | 0 | 0 | 0 | 0.28 ± 0.09 [c] | 0.51 ± 0.14 [b] | 0.77 ± 0.14 [a] | 0.95 ± 0.13 [a] |
| | 0 | 1.27 | 0 | 0 | 1.82 ± 0.21 [c] | 2.33 ± 0.20 [a,b] | 2.92 ± 0.23 [a] | 3.03 ± 0.21 [a] |
| MS | 1.0 | 0 | 0 | 1.0 | 2.67 ± 0.12 [c] | 2.82 ± 0.17 [a,b] | 3.26 ± 0.23 [a] | 4.51 ± 0.09 [a] |
| | 1.5 | 0 | 0 | 1.5 | 3.13 ± 0.08 [c] | 3.33 ± 0.21 [ab] | 4.33 ± 0.13 [a] | 4.79 ± 0.29 [a] |
| | 2.0 | 0 | 0 | 2.0 | 1.41 ± 0.18 [c,d] | 3.85 ± 0.22 [a] | 3.41 ± 0.23 [a,b] | 4.64 ± 0.21 [a] |

* Different lowercase letters in the same column indicated the significant difference at $p \leq 0.05$ (Duncan's multiple-range test).

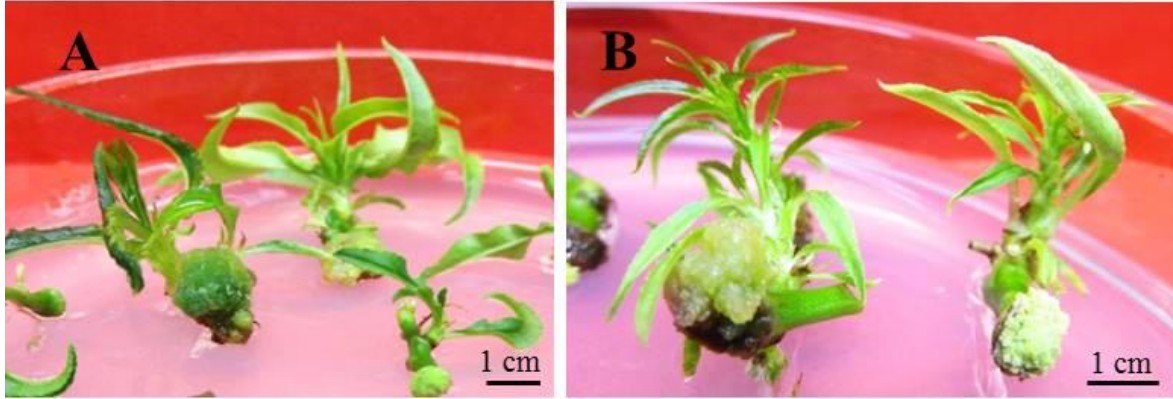

**Figure 3.** Callus formation at the base of explants in peach hybrids: (**A**) 'Loadel' × 'Nikitskiy Podarok' on MS medium with 1.27 mg L$^{-1}$ TDZ; (**B**) 'Jerseyglo' × 'Nikitskiy Podarok' on MS medium with 1.5 mg L$^{-1}$ BAP + 1.5 mg L$^{-1}$ IAA.

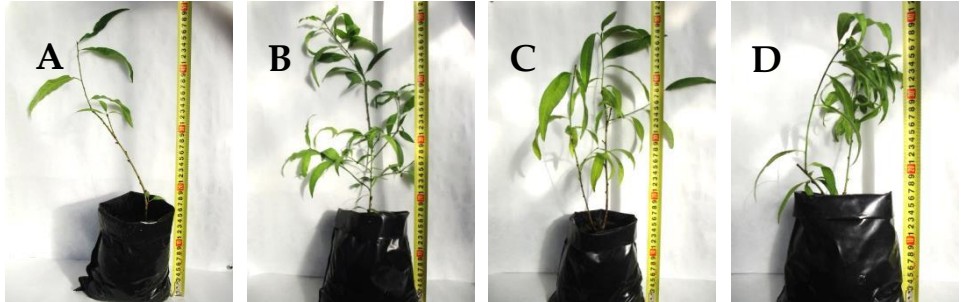

**Figure 4.** Adapted peach seedlings: (**A**) 'Clyde Wilson' × 'Nikitskiy Podarok'; (**B**) 'Jerseyglo' × 'Nikitskiy Podarok'; (**C**) 'Loadel' × 'Nikitskiy Podarok'; (**D**) 'Summerglo' × 'Nikitskiy Podarok'.

*3.4. Leaf Micromorphological and Anatomical Characteristics*

The leaves of the studied peach hybrids were bifacial, hypostomatic, their thickness varied from 97 to 127 µm (under different culture conditions), covered with a cuticle on both the adaxial and abaxial sides (Table 4). They had reticular venation. The cuticle

thickness ranged from 2 to 8 μm. The vascular bundle was of a closed collateral type, and had the shape of a regular arc (Figure 5(1), Figure 6(1) and Figure 7(1)). In the initial period of post aseptic adaptation (15 days ex vitro), there was an increase in the number of vessels in the secondary xylem, the appearance of the sclerenchymal sheath and collenchyma. The leaf petiole in the cross section was of a rounded triangular shape. The bark of the bundle was thin, its two outer layers were represented by collenchyma. Simple trichomes were found on the abaxial epidermis under the central vein.

**Table 4.** Structural characteristics of leaves in peach hybrids under in vitro culture, and acclimatization of plantlets ex vitro and in vivo.

| Parameters | Cross Combinations | In Vitro * | Ex Vitro * | Iv Vivo * |
|---|---|---|---|---|
| | | | 15 Days | 60 Days |
| Leaf blade thickness, μm | 'Clyde Wilson' × 'Nikitskiy Podarok' | 97.2 ± 2.4 [d,e] | 100.3 ± 3.3 [b,c] | 105.5 ± 1.8 [a,b] |
| | 'Jerseyglo' × 'Nikitskiy Podarok' | 103.3 ± 1.3 [b,c] | 113.5 ± 2.2 [ab] | 119.0 ± 1.4 [a] |
| | 'Loadel' × 'Nikitskiy Podarok' | 98.7 ± 2.6 [c,d] | 114.3 ± 2.5 [ab] | 127.2 ± 2.7 [a] |
| | 'Summerglo' × 'Nikitskiy Podarok' | 100.6 ± 1.4 [b,c] | 108.5 ± 1.2 [ab] | 116.3 ± 3.0 [a] |
| Palisade mesophyll/thickness, μm | 'Clyde Wilson' × 'Nikitskiy Podarok' | 51.3 ± 1.8 [f,g] | 51.6 ± 2.0 [e,f] | 52.3 ± 1.3 [c,d] |
| | 'Jerseyglo' × 'Nikitskiy Podarok' | 59.5 ± 1.1 [a,b] | 63.5 ± 1.3 [a,b] | 58.3 ± 1.2 [a,b] |
| | 'Loadel' × 'Nikitskiy Podarok' | 51.6 ± 0.8 [e,f] | 56.3 ± 2.4 [b,c] | 69.0 ± 1.5 [a] |
| | 'Summerglo' × 'Nikitskiy Podarok' | 52.4 ± 1.4 [c,d] | 53.9 ± 1.3 [b,c] | 58.3 ± 1.1 [a,b] |
| Adaxial epidermis thickness, μm | 'Clyde Wilson' × 'Nikitskiy Podarok' | 13.5 ± 0.6 [c,d] | 14.6 ± 0.9 [bc] | 14.9 ± 0.7 [a,b] |
| | 'Jerseyglo' × 'Nikitskiy Podarok' | 13.6 ± 0.7 [c,d] | 16.1 ± 0.7 [ab] | 16.9 ± 0.7 [a,b] |
| | 'Loadel' × 'Nikitskiy Podarok' | 15.2 ± 0.6 [a,b] | 18.1 ± 0.5 [a,b] | 18.8 ± 0.7 [a] |
| | 'Summerglo' × 'Nikitskiy Podarok' | 18.1 ± 0.6 [a,b] | 20.0 ± 0.4 [a] | 19.5 ± 0.8 [a] |
| Abaxial epidermis thickness, μm | 'Clyde Wilson' × 'Nikitskiy Podarok' | 7.5 ± 0.3 [f,g] | 8.7 ± 0.4 [d,e] | 11.7 ± 0.7 [c,d] |
| | 'Jerseyglo' × 'Nikitskiy Podarok' | 16.9 ± 0.5 [a] | 19.1 ± 0.4 [a] | 17.1 ± 0.7 [a] |
| | 'Loadel' × 'Nikitskiy Podarok' | 12.1 ± 0.5 [b,c] | 14.3 ± 0.6 [a,b] | 15.1 ± 0.7 [a,b] |
| | 'Summerglo' × 'Nikitskiy Podarok' | 12.4 ± 0.5 [b] | 14.7 ± 0.6 [a,b] | 14.1 ± 0.7 [a,b] |
| Stomata number per 1 mm² surface on the abaxial leaf side | 'Clyde Wilson' × 'Nikitskiy Podarok' | 157 ± 5 [a,b] | 95 ± 3 [c,d] | 139 ± 2 [a,b] |
| | 'Jerseyglo' × 'Nikitskiy Podarok' | 115 ± 4 [b,c] | 85 ± 4 [f,g] | 159 ± 2 [a,b] |
| | 'Loadel' × 'Nikitskiy Podarok' | 182 ± 5 [a] | 95 ± 2 [c,d] | 147 ± 3 [a,b] |
| | 'Summerglo' × 'Nikitskiy Podarok' | 207 ± 4 [a] | 93 ± 2 [d,e] | 133 ± 4 [a,b] |

* Different lowercase letters in the same column indicated the significant difference at $p \leq 0.05$ (Duncan's multiple-range test).

Mesophyll in the hybrids 'Jerseyglo' × 'Nikitskiy Podarok', 'Loadel' × 'Nikitskiy Podarok' and 'Summerglo' × 'Nikitskiy Podarok' was dense, of 5–6 layers, while the hybrid 'Clyde Wilson' × 'Nikitskiy Podarok' had 4–5 layers (Figures 5(2)–7(2)). Cases of isopalisade mesophyll formation were observed in the hybrid 'Summerglo' × 'Nikitskiy Podarok' under in vitro conditions and 60 days adaptation in vivo. The loosest arrangement of mesophyll cells at all culture stages was observed in the hybrid 'Clyde Wilson' × 'Nikitskiy Podarok' (Figures 5–7A). Palisade tissue had 2 cell layers, spongy tissue-3–4.

Epidermis was single-layered, cell thickness was 13–20 μm on the adaxial side and 7–19 μm on the abaxial one. Cells were 5–7-angular in shape, elongated, large on the adaxial leaf surface and small, elongated on the abaxial surface (Figures 5(3,4)–7(3,4)).

Stomatal apparatus were of the anomocytic type. Their number varied depending on the culture and the adaptation period conditions. Thus, the maximum number of stomata was recorded in vitro (115 to 207 stomata mm²), the length of the stomata pore was 10–24 μm. The largest stomata were found in the hybrids 'Clyde Wilson' × 'Nikitskiy Podarok' and 'Loadel' × 'Nikitskiy Podarok', the smallest ones–in 'Jerseyglo' × 'Nikitskiy Podarok' (Figure 5(4)).

Most of the stomata were open. Under 15 days adaptation within, the number of stomata in all genotypes decreased to 85–95 stomata per mm², their size was 13–24 μm. Moreover, most of the stomata were closed. A relatively even stomata distribution in

all hybrid forms was observed under 60 days adaptation, when the cells of the abaxial epidermis became small (21–34 µm long, 9–16 µm wide), the number of stomata increased (Table 4).

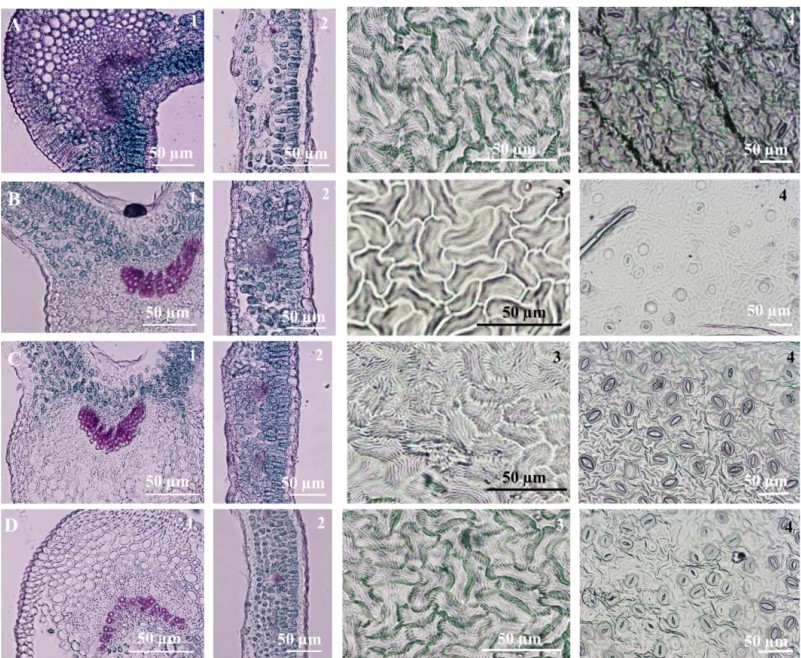

**Figure 5.** Cross sections of leaves in peach hybrids cultured in vitro: (**1**) central vein, (**2**) leaf, and casts of their integumentary tissues (**3**) adaxial epidermis, (**4**) abaxial epidermis. (**A**) 'Clyde Wilson' × 'Nikitskiy Podarok', (**B**) 'Jerseyglo' × 'Nikitskiy Podarok', (**C**) 'Loadel' × 'Nikitskiy Podarok', (**D**) 'Summerglo' × 'Nikitskiy Podarok'. Bars = 50 µm.

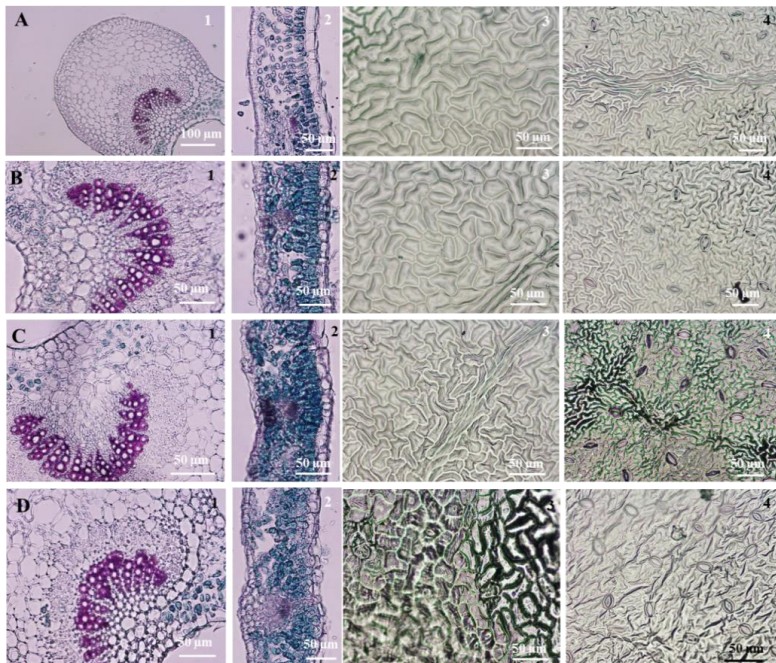

**Figure 6.** Cross sections of leaves in peach hybrids under ex vitro acclimatization for 15 days: (**1**) central vein, (**2**) leaf and casts of their integumentary tissues (**3**) adaxial epidermis, (**4**) abaxial epidermis). (**A**) 'Clyde Wilson' × 'Nikitskiy Podarok', (**B**) 'Jerseyglo' × 'Nikitskiy Podarok', (**C**) 'Loadel' × 'Nikitskiy Podarok', (**D**) 'Summerglo' × 'Nikitskiy Podarok'. Bars = 50 µm.

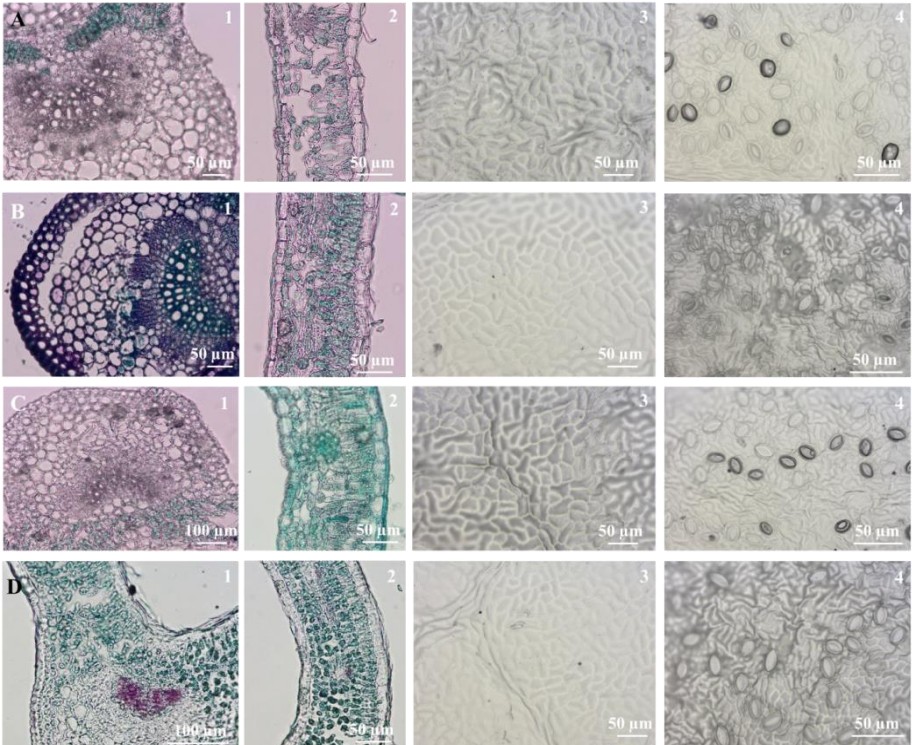

**Figure 7.** Cross sections of leaves in peach hybrids under in vivo acclimatization for 60 days: (**1**) central vein, (**2**) leaf and casts of their integumentary tissues (**3**) adaxial epidermis, (**4**) abaxial epidermis. (**A**) 'Clyde Wilson' × 'Nikitskiy Podarok', (**B**) 'Jerseyglo' × 'Nikitskiy Podarok', (**C**) 'Loadel' × 'Nikitskiy Podarok', (**D**) 'Summerglo' × 'Nikitskiy Podarok'. Bars = 50 μm.

### 3.5. Functional State of Peach Hybrids In Vitro and during the Adaptation In Vivo

The obtained results demonstrated a gradual increase in chlorophyll amount from 12–26 μg cm$^{-2}$ in vitro to 43–50 μg cm$^{-2}$ in the adapted plants (Figure 4A). At the same time, in the hybrid 'Clyde Wilson' × 'Nikitskiy Podarok' chlorophyll amount was less than in other forms, which is most likely associated with the structural features of its leaves. The maximum values of this parameter were recorded in 'Loadel' × 'Nikitskiy Podarok' plants (except for the 15–day adaptation period).

Chlorophyll fluorescence induction data indicated relatively low photosynthetic activity in vitro in all studied cultivars–from 0.36 to 0.46 a.u. (Figure 8B), while the efficiency of photosynthesis was quite high (0.28–0.40 a.u.), especially in the hybrid 'Jerseyglo' × 'Nikitskiy Podarok' (Figure 8C). After 15 days adaptation, the activity of assimilation processes increased, and by the 60-day culture period in vivo it was 0.68–0.72 a.u. with a photosynthetic efficiency of 0.42–0.54 a.u.

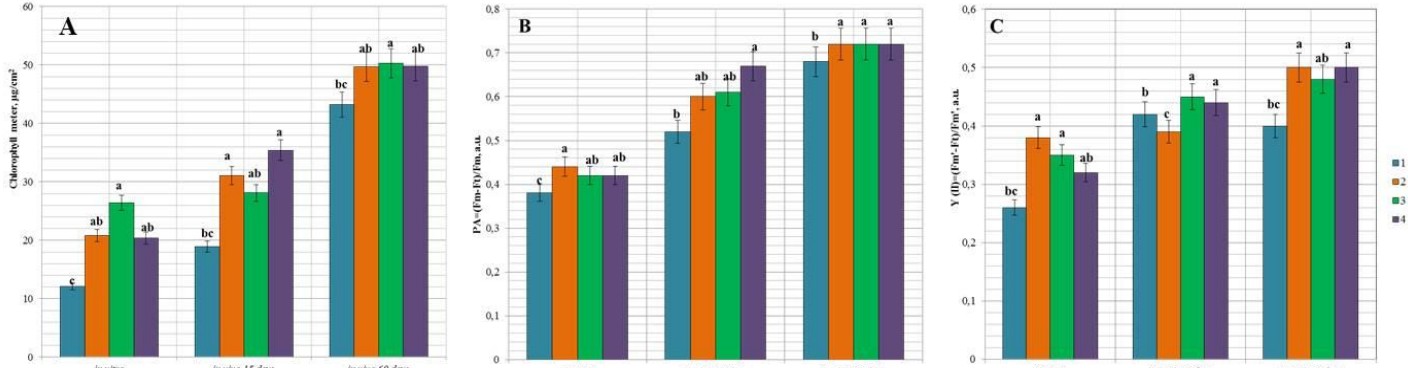

**Figure 8.** Parameters of the assimilation apparatus functioning in peach hybrids in vitro and under acclimatization in vivo: (**1**) 'Clyde Wilson' × 'Nikitskiy Podarok', (**2**) 'Jerseyglo' × 'Nikitskiy Podarok', (**3**) 'Loadel' × 'Nikitskiy Podarok', (**4**) 'Summerglo' × 'Nikitskiy Podarok'. (**A**) chlorophyll meter, $\mu$g/cm$^2$, (**B**) photosynthetic activity, (**C**) effective photochemical quantum yield of PS II. Different lowercase letters in the same column indicated the significant difference at $p \leq 0.05$ (Duncan's multiple-range test).

## 4. Discussion

Our studies showed different morphogenetic responses under in vitro culture of peach hybrid embryos of four cross combinations. Many researchers have reported that the frequency of plantlet in vitro regeneration depends on the effect of various factors [40–47]. The ability of peach and other stone fruit crops embryos to form normal plantlets depends, to a great extent, on the exposure to a low positive temperature of 4–5 °C without light [30,48–50]. In our experiments, the absence of cold pretreatment resulted in abnormal development of seedlings in all cross combinations with undeveloped roots. Daorden et al. [51] have reported that stratification and in vitro culture made it possible to shorten the time of peach embryo germination and obtain 91% of developed plants from seeds stratified at 4 °C. At the same time, for cultivated peach ovules, using 0.5 °C and the addition of 6% and 8% sucrose to the culture medium gave the positive effect for the embryo development [52].

Embryo size, culture medium, PGRs have a great effect on the normal development of embryos in vitro. Thus, in our experiments, embryos less than 1.0 cm long on hormone-free Monnier culture medium did not develop or developed poorly and formed single defective seedlings. On this culture medium the lowest frequency of shoot regeneration from hybrid embryos 0.3–1.0 cm long in 'Clyde Wilson' × 'Nikitskiy Podarok' (3.70%) and 'Jerseyglo' × 'Nikitskiy Podarok' (7.41%) was noted. The regeneration frequency of embryos 'Summerglo' × 'Nikitskiy Podarok' and 'Loadel' × 'Nikitskiy Podarok' was not more then 14.81 and 18.51, respectively. Ramming [52] showed better development of embryos more than 1.0 cm long on SBH culture medium, for embryos less than 1.0 cm long on MS medium with 1240 mg L$^{-1}$ potassium succinate and 584.6 mg L$^{-1}$ L–glutamine was needed. Many researchers have noted the need for complex BAP with GA$_3$ or BAP with IBA in the culture medium [53–55]. Şan et al. [56] and Ghayyad [49] have also reported about the germination of embryos isolated from cotyledons in stone fruit crops, including peach, on culture media with various BAP and GA$_3$ concentrations. Nagaty [57] obtained a high rate of shoot regeneration from mature embryos of Taif peach cultivar when incubated in the dark for the first 13 days of culture on the medium with 3.6 μM TDZ and 2.5 μM IBA. Srivastav et al. [55] showed that only mature embryos (60 days after pollination) should be used to obtain viable early maturing subtropical peach forms. In our studies for future regeneration with embryos less than 1.0 cm in size we used 0.4 mg L$^{-1}$ kinetin and 0.15 mg L$^{-1}$ GA$_3$ in Monnier's culture medium that induced germination and formation of shoots and roots. However, the frequency of seedling regeneration did not exceed 29%. Mature seedlings were obtained from all peach hybrid embryos 1.1–2.0 in size on hormone-free Monnier medium with 400.0 mg L$^{-1}$ casein hydrolyzate. Hybrid embryos

'Summerglo' × 'Nikitskiy Podarok' and 'Loadel' × 'Nikitskiy Podarok' were characterized by high regeneration capacity (96.30 and 92.59%, respectively). These data correspond to those by Monnier [58], Bridgen [59] and Devi et al. [50] that in the mature embryos culture, PGRs are usually not used or are used in low concentrations, since high concentrations of PGRs reduce the germination rate of peach embryos.

Some obtained seedlings of four peach cross combinations for morphogenic capacity investigation were used. It was demonstrated that B5 medium with 0.75 mg $L^{-1}$ BAP and 0.1 mg $L^{-1}$ IBA was optimal for four cross combinations of peach in number of newly adventitious shoots formation (from three to five shoots per explant).

The leaf blades of the studied hybrid forms had a complex of common anatomical and morphological features typical for *P. persica* crop: the dorsoventral hypostomatic type of the leaf blade, the vascular system of the closed collateral type, dense mesophyll, developed cuticular integuments, the presence of cover trichomes along the central vein [60,61]. These authors also have pointed out that under open field conditions, leaf thickness in different peach genotypes could be 84–142 µm, and the number of mesophyll layers varied from five to eight [60,62]. Isopalisad mesophyll was found in drought–tolerant cultivars. Thus, under in vitro culture, all the studied hybrid forms had structural features typical for ex situ peach plants. The exception was the density of the palisade and spongy tissue layers and the number of these layers. However, during ex vitro acclimatization time, the thickness of the leaf and its tissue increased.

Similar results for the leaf structure of in vitro micropropagated and in vivo adapted plants were obtained by Calvete et al. [63] in *Fragaria ananassa* Duch. (Rosaceae) plants, Romano and Martins-Loucao [12] in the culture of *Quercus ruber* L. (Fagaceae), Yang and Yeh [64] in *Calatheaor bifolia* (Linden) H. Kenn. (Marantaceae), Batagin-Piotto et al. [65] in the culture of *Bactris gasipaes* (Arecaceae), Werner et al. [14] in the culture of *Crambe abyssinica* Hochst (Brassicaceae), I. Mitrofanova et al. [66] in *Canna indica* L. (Cannaceae), O. Mitrofanova et al. [15] in *Lavandula angustifolia* Miller. and *L. × intermedia* Emeric ex Loisel. (Lamiaceae), Copetta et al. [26] in *Mertensia maritima* (L.) Gray (Boraginaceae). The authors have identified features typical for in vitro leaves in comparison with in vivo plants: less mesophyll differentiation, smaller palisade thickness, and development of large intercellular spaces.

Sarikhani and Sarikhani-Khorami [67], examining the leaves of the hybrid *P. persica* × *P. davidiana* microshoots in vitro, indicated a high correlation between stomata formation and control of their activity and different types of light intensity. Thus, at a high intensity of red light, fewer stomatal apparatuses are formed, and the thickness of the leaf and its structural layers are thinner. The combined red and blue light induced more chlorophyll and larger stomata. In our work, we used 37.5 µmol $m^{-2}$ $s^{-1}$ white luminescent lamps, which can be realized in developing of leaves characterized with a significant thickness and multiple formation of stomatal apparatus.

Our observations showed that, with a comparatively similar leaf thickness of the studied hybrid forms, both in vitro and under gradual in vivo adaptation, differences were noted in the thickness of the epidermis, palisade mesophyll, and the size of the intercellular spaces of the mesophyll. Thick epidermis and palisade parenchyma can increase resistance to water stress and ensure growth under conditions of acclimatization, improving the water regime and protection of leaf tissues [68,69]. The densest mesophyll was noted in the hybrids 'Loadel' × 'Nikitskiy Podarok' and 'Jerseyglo' × 'Nikitskiy Podarok'. The intercellular spaces in the spongy mesophyll did not exceed 10%. These genotypes were characterized by higher ratio of palisade to spongy mesophyll that suggests a more compact arrangement of cells and a larger surface area of the mesophyll per leaf unit. In turn, these features can promote $CO_2$ uptake and thus support photosynthesis under water stress during acclimatization [70]. Some researchers have concluded that the compact arrangement of the palisade mesophyll layers results in increased mechanical strength of the parenchyma tissue and protects the leaves from the excess water loss [60,68,69].

The hybrids 'Summerglo' × 'Nikitskiy Podarok' and 'Loadel' × 'Nikitskiy Podarok' were characterized by the maximum thickness of the adaxial epidermis, while the cross combination 'Jerseyglo' × 'Nikitskiy Podarok' had the maximum thickness of the abaxial epidermis. Palasciano et al. [71] and Camposeo et al. [72] reported in their works that cultivars with fewer stomata (or their low frequency) may represent a model of acclimatized genotypes. Stomata distribution indicates a better ability to regulate gas exchange in vitro and at the initial stages of acclimatization in 'Jerseyglo' × 'Nikitskiy Podarok' plants, and after 60 days of post aseptic adaptation–in 'Summerglo' × 'Nikitskiy Podarok'.

Previously, growth and physiological changes in some plant species under in vitro propagation have been described [27,28]. Insufficient work of the photosynthetic apparatus is associated with the addition of sucrose to the culture media and $CO_2$ deficiency [73,74]. Photoinhibition in peach plants and changes in the general functional state under the pressure of abiotic stress factors have been shown [75–77]. In our work, low photosynthetic activity and relatively low values of photosynthesis Y (II) efficiency were associated with a low chlorophyll amount in vitro. This may be a result of an increase in the chlorophyll b content in the a/b ratio, which indicates the high concentration of light-harvesting complexes [78]. Some authors have pointed to a decrease in photosynthetic activity during the initial stages of adaptation [73,79]; however, in our work, depression of photosynthetic processes at 15 days adaptation was not observed.

Chlorophylls are the main light-absorbing pigments and key components of photosynthesis in plants. Physiological studies of peach have revealed that plants with low resistance to water stress had a significantly lower chlorophyll amount [80]. A slight increase in the amount of chlorophyll during the 15–day adaptation period, along with an increase in assimilation parameters, indicates successful processes of post aseptic acclimatization. The subsequent growth of $F_v/F_m$ value is comparable to those of ex situ cultivated peach plants [78]. Effective activity of photosystem II during a 60–days in vivo culture lead us to conclude that the leaf apparatus of peach hybrid forms functioned normally, the necessary phytomass was accumulated, and the seedlings could be planted in an open field. The minimum values of photosynthetic activity and efficiency of photosystem II were noted for the hybrid 'Clyde Wilson' × 'Nikitskiy Podarok'.

## 5. Conclusions

The main factors (embryo size, stratification, culture medium, RGRs) were allowed to develop normal plantlets from hybrid embryos of four peach cross combinations. Mature seedlings were obtained from all peach hybrids with embryos 1.1–2.0 cm by size on hormone-free Monnier culture medium. The regeneration capacity in cross-combinations 'Summerglo' × 'Nikitskiy Podarok' and 'Loadel' × 'Nikitskiy Podarok' reached 96.30 and 92.59%, respectively. Using micropropagation technique induced adventitious shoots regeneration from nodal explants of undeveloped peach hybrid seedlings on B5 medium with 0.75 mg L$^{-1}$ BAP + 0.1 mg L$^{-1}$ IBA. The leaf blades in the peach hybrids obtained in vitro have mature integumentary, mechanical and assimilating tissues, actively transpire and efficiently use light energy for photosynthesis. Their thickness was 97–103 μm. Acclimatization is gradual, and after 60 days in vivo in the hybrids 'Jerseyglo' × 'Nikitskiy Podarok', 'Loadel' × 'Nikitskiy Podarok' and 'Summerglo' × 'Nikitskiy Podarok', significant closure of chlorenchyma cells, the presence of a thick cuticle and trichomes were noted. These features are the evidence of the xeromorphic structure of the leaves and they are typical for drought-tolerant cultivars ex situ. The newly formed leaves in all studied genotypes actively photosynthesize (($F_m − F_t$) / $F_m$ = 0.68 − 0.72 a.u.) and efficiently use the obtained energy for assimilation processes (Y (II) = 0.42–0.54 a.u.). There were no photoinhibition processes, so we may suppose the normal functioning of peach hybrid plants at all stages of plantlets development from embryos.

This protocol of plantlets obtaining from hybrid embryos is appropriate for the peach breeding programs on creation of new economically valuable cultivars. Additionally,

this investigation may be helpful for subsequent studies of peach on virus resistance or tolerance.

**Author Contributions:** Conceptualization, I.M.; methodology, I.M., N.L.-S., V.T. and O.M.; formal analysis, I.M., N.L.-S. and V.T.; investigation, I.M., N.L.-S., V.T. and O.M.; resources I.M. and A.S.; software I.M. and O.M.; project administration I.M.; data curation, I.M., N.L.-S. and V.T.; writing— original draft, I.M., N.L.-S. and V.T.; writing—review and editing, I.M., N.L.-S., V.T. and O.M.; funding acquisition, I.M. and A.S. All authors have read and agreed to the published version of the manuscript.

**Funding:** This research was funded by the Russian Science Foundation, grant number 19-16-00091.

**Institutional Review Board Statement:** Not applicable.

**Informed Consent Statement:** Not applicable.

**Data Availability Statement:** Raw data are available on request from the corresponding author.

**Conflicts of Interest:** The authors declare no conflict of interest.

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
