# Peer review of "Use of Biotechnological Methods to Support the Production of New Peach Hybrids"

_horticulturae, doi:10.3390/horticulturae7120533_

Round 1

Reviewer 1 Report

they shoud indicate if the plants regenerated  form the embryos could be mircropropagated. The agronomical interest of the selected  cultivars used in this work   must be better explained in the introduction.

Author Response

Dear Reviewer, 

Please see the attachment with my answers and comments.

Kind regards.

Reviewer 2 Report

Review on Mitrofanova et al.  Creation of New Promising Peach Breeding Forms With the Use of Biotechnological Methods:

The comparative study reports on the in vitro cultured peach plantlets and their regeneration, that are coming from different crosses. Although the measured parameters may lead to the improvement of in vitro culturing protocols of peach and the text is in general in a good shape (with relatively few mistakes), the experimental design and the objectives of the study in its present form is incoherent, and therefore I cannot propose acceptance for publication.

The title is misleading: „Creation of New Promising Peach Breeding Forms With the Use of Biotechnological Methods”. It is not clear at all what does the word “promising” mean here (later it is explained that creating peach hybrids with elevated resistance against virus induced diseases are the objective of this study), but nothing is known about the source of putative virus resistance in hybrids. What are the reasons of choosing those parental genotypes that had been chosen? What are the perspective phenotypic characteristics of each of the parental genotypes?

Virus-free propagation material had been used for the creation of in vitro cultures and no virus infection or any other studies had been conducted to see if they are in fact resistant to viruses. It is not clear how the biotechnology used for their in vitro propagation is related to their perspectivity? Why biotechnology was necessary to apply to have new virus tolerant genotypes (if they are so)?

In discussion, a more coherent description should be involved where the authors evaluate the differences among the hybrids with different genetic backgrounds in terms of all the measured parameters.

The plantlets are not hybrid „forms”, they are hybrids

L140: under greenhouse conditions

In vitro, ex vitro and other foreign expressions should be italicized

„hybrid forms embryos” expression is of bad grammar, instead of it probably hybrid embryos

Lines 447-455: I wonder if such a great number of species and studies are all relevant to this study.

L474: The grammar is incorrect: „were characterized by high the ratio of palisade mesophyll to spongy…”

L510-511: and what does it mean?

Author Response

(The authors gave the same response as above.)

Reviewer 3 Report

Dear Authors

first I should say thank you for your concern in resolving a problem in the peach industry but it would be nice to have a better-described experiment design in the manuscript. you said that the objective of your study was to determine the tolerance of different genotypes against viral infections but you did not share how with a simple embryo culture technique without any virus inoculation you want to measure that? I am sure you can help readers have a better understanding of your total scientific journey on this subject.

yours

Author Response

Dear Reviewer,

Please see the attachment with our answers and comments.

Kind regards.
